

# A cross-sectional questionnaire study of the rules governing pupils' carriage of inhalers for asthma treatment in secondary schools in North East England

Wendy Funston and Simon J. Howard

Health Education North East, Newcastle upon Tyne, United Kingdom

## ABSTRACT

**Objectives.** The primary objective of this study was to assess the rules governing secondary school pupils' carriage of inhalers for emergency treatment of asthma in the North East of England.

**Design.** This study was based upon a postal questionnaire survey.

**Setting.** The setting for this study was mainstream free-to-attend secondary schools which admit 16 year old pupils within the 12 Local Authority areas which make up the North East of England.

**Participants.** All 153 schools meeting the inclusion criteria were invited to participate in the study, of which 106 (69%) took part.

**Main Outcome Measures.** Our three main outcome measures were: whether pupils are permitted to carry inhalers on their person while at school; whether advance permission is required for pupils to carry inhalers, and from whom; and whether the school has an emergency 'standby' salbutamol inhaler for use in asthma emergencies, as permitted since October 2014 under recent amendments to The Human Medicines Regulations 2012.

**Results.** Of 98 schools submitting valid responses to the question, 99% ($n = 97$) permitted pupils to carry inhalers on their person while at school; the remaining school stored pupils' inhalers in a central location within the school. A total of 22% of included schools ($n = 22$) required parental permission before pupils were permitted to carry inhalers. Of 102 schools submitting valid responses to the question, 44% ($n = 45$) had purchased a 'standby' salbutamol inhaler for use in asthma emergencies.

**Conclusions.** Most secondary schools in North East England permit pupils to carry inhalers on their person. The requirement in a minority of schools for parental permission to be given possibly contravenes the standard ethical practices in clinical medicine for children of this age. Only a minority of schools hold a 'standby' salbutamol inhaler for use in asthma emergencies. Wider availability may improve outcomes for asthma emergencies occurring in schools.

Corresponding author
Simon J. Howard,
sjhoward@doctors.org.uk

## INTRODUCTION

Asthma is the most common long-term condition in childhood in the UK (*NICE, 2013*), with 1.1 million children currently receiving treatment (*NICE, 2013*). The National Review of Asthma Deaths (*Levy et al., 2014*) found that 14% of all asthma deaths in the UK occurred in those aged under 20 years, and 75% of asthma deaths in this age group occurred outside of hospital. In addition to a high burden of mortality, asthma accounts for large numbers of hospital admissions: in the twelve months to May 2014, there were 54,300 emergency hospital admissions for asthma, of which 20,510 (38%) were in children (*Health and Social Care Information Centre, 2014*). Up to 70% of emergency asthma admissions may be preventable with appropriate early interventions (*NICE, 2013*).

The ability to improve outcomes in young people who suffer acute exacerbations of asthma in the community depends on high-quality emergency care in settings such as schools, where young people spend a large proportion of their time. Indeed, Asthma UK estimates that two thirds of children with asthma aged 5–18 years have experienced an asthma attack at school (*Asthma, 2013*).

The unpredictable nature of asthma requires that those aiming to deliver high-quality community care recognise acute exacerbations and promptly initiate acute medical management. In most cases, initial medical management will include use of a short acting bronchodilator delivered via an inhaler. Hence, if the exacerbation is to be controlled, rapid access to inhaler medication is a prerequisite.

Some variation in the method of achieving ready access to inhaler medication is to be expected since access will be affected by the wider context of school layout and competing policies. However, for older children, a simple method of facilitating ready access is to allow pupils to carry inhaler medication with them. The age at which a child achieves the maturity necessary to responsibly carry their own medication will vary from child to child. In England and Wales, under section 8 of the Family Law Reform Act 1969 (*Government of the United Kingdom, 1969*), those aged 16 years are presumed to be capable of consenting to their own medical treatment and any ancillary procedures involved in that treatment (though refusal of treatment may, in certain circumstances, be overridden by a person with parental responsibility or a court of law). As a result, it is probably reasonable for schools to presume that children aged 16 years are responsible enough to carry their own emergency asthma inhalers.

Under statutory guidance introduced in 2014 (*Department for Education, 2014*), schools are responsible for ensuring that appropriate policies are in place to support children with medical conditions. However, anecdotal evidence suggests that schools have adopted a variety of approaches to supporting children with asthma. In particular, the rules regarding storage of inhalers appear to vary, ranging from inhalers being locked in a central location to pupils being permitted to keep inhalers in their pockets. Location of reliever medication forms a key part of its level of accessibility in an emergency situation. Indeed, there have been a number of media reports of children in the UK (*Buckingham, 2014*; *Farr, 2010*) and elsewhere (*The Canadian Press, 2013*; *Chaiyanhat, 2012*) dying or becoming seriously unwell as a result of an asthma attack in school, with reports citing a lack of ready access

to inhaler medication as a contributory factor. However, there is little published data describing the variation in practice. Having a better understanding of the rules governing carriage of inhalers in schools may highlight areas in which asthma management can be improved for young people.

As a result of extensive campaigning on the issue of inhaler availability in schools, the UK Government laid legislation before Parliament in July 2014 (*Government of the United Kingdom, 2014*) to amend The Human Medicines Regulations 2012 (*Government of the United Kingdom, 2012*) to allow schools to purchase emergency salbutamol inhalers. Such inhalers are for the use of children diagnosed with asthma and prescribed an inhaler, where parents have given written permission for the emergency inhaler to be used. The legislation (*Government of the United Kingdom, 2014*) permitted schools to purchase emergency inhalers from 1 October 2014. This regulatory change may represent a useful step in facilitating access to emergency asthma treatment in schools; however, there is no published data on the number of schools who have availed of this new power, and so no assessment of the impact of the legislation is possible.

This study aims to contribute data to two substantial knowledge gaps in asthma policy. Firstly, we assess the rules governing carriage of inhalers in secondary schools in North East England in aiming to gain a better appreciation of current practice. Secondly, we assess the proportion of secondary schools in North East England which have taken the opportunity to purchase an emergency salbutamol inhaler.

## METHODS

We compiled a list of all free-to-attend schools within the 12 Local Authorities in North East England using listings on Local Authority websites between October 2014 and February 2015. We reviewed the website belonging to each school, or (when no accessible website could be located) the latest Ofsted report. Schools whose website or Ofsted report indicated that they served 16-year-old mainstream pupils were included in our sample population.

We limited our study to schools which served 16-year-old mainstream pupils in order to aid interpretation of our results. Given the presumption of capacity to accept medical treatment in this age group, we felt that it was reasonable to presume capacity to carry inhalers in mainstream pupils in this age group. In addition, we excluded schools serving only pupils with special educational needs and educational institutions other than schools (such as Local Authority 'pupil referral units' for children with behavioral difficulties). Without these restrictions applied, results would be difficult to interpret: a policy allowing pupils to carry their own inhalers in a school catering only for young children or those with learning or behavioral difficulties may be considered irresponsible. We also excluded fee-paying schools as we were unable to locate a complete listing of these.

We undertook an initial pilot study in one Local Authority. Where available, the school policy governing inhaler carriage was accessed via the school website. As few schools had an available online policy, we sent emails to invite the remainder to supply their policy by email. As few schools responded to emails, we sent postal letters to invite the remainder to complete a brief online or postal questionnaire describing their school policy. Each questionnaire response was considered and categorised by both authors.

Since postal letters resulted in the highest response rate in the pilot phase, all included schools in the 11 remaining Local Authorities were invited by postal letter to complete an online or postal questionnaire. Postal letters were addressed to the head teacher of each school. The questionnaire consisted of two questions. The first question asked the responder to select a response describing the school's policy regarding storage or carriage of inhalers for asthma. A free text box was also provided with this question to enable responders to describe any other arrangements for carriage of inhalers that did not feature in the prepared list of options. The second question asked if the school had an emergency salbutamol inhaler for use by any pupil in an asthma emergency. The questionnaire did not specify any age group of pupils to which the policies applied. A copy of the questionnaire is available as a File S1.

On receipt of the questionnaires, both authors considered the responses and entered the data into a central database. Discussion was used to reach a consensus where responses were not clear cut (such as where multiple responses were selected, and had to be considered in conjunction with clarifying free text). Responses which the authors felt unable to reliably interpret (for example, where multiple mutually exclusive responses were given) were categorised as invalid.

Data was collected during the period November 2014–May 2015.

Ethical approval was not required for this research as it was based solely on collation of policy information.

## RESULTS

Our review of 12 Local Authority websites found 1,133 free-to-attend schools for consideration. The number of schools per Local Authority ranged from 35 to 275 (mean 94.4). On review of the schools' websites, 980 of these (86%) were excluded from further consideration in this study. The majority of these ($n = 915$, 93%) were excluded as the schools did not accept pupils aged 16 years. The remaining exclusions were due to: the school serving only pupils with special educational needs ($n = 55$, 6%); the institution being an educational institution other than a school, such as a Local Authority pupil referral unit ($n = 9$, 1%); or the school being a fee-paying private school mis-categorised on the Local Authority website ($n = 1$, 0%). Hence, the policies of 153 schools were included in this study, with the number per Local Authority ranging from 5 to 31 (mean 12.7).

Of 153 schools, we received responses from 106 (69%). 100 (94%) of these responses were received as postal questionnaire responses; 3 (3%) as electronic questionnaire responses; and 3 (3%) were answers derived by the authors from reviewing school policies. The response rate varied between Local Authorities, ranging from 43% to 100%.

The first question concerned the rules governing pupils' carriage of inhalers at school. Of the 106 responses received to this question, we excluded 8 (8%) due to a lack of clarity: for example, selecting two or more mutually exclusive responses, or adding a free-text comment which was inconsistent with the selected multiple choice response. Hence, 98 responses were included in this part of the study.

Of these 98 responses, 75 (77%) indicated that children were automatically allowed to carry their own inhalers at school. 22 (22%) indicated that children were only allowed to

carry their inhalers with parental consent. The remaining school (1%) indicated that pupils were not allowed to carry inhalers, but that inhalers were stored in a single central location within the school.

The second question concerned whether schools had a generic salbutamol inhaler for use by any pupil in an asthmatic emergency. We excluded the three schools whose response to the first question had been derived from policies since the policies were unclear in this area. We additionally excluded one questionnaire response due to a lack of clarity. Hence, 102 questionnaire responses were included in this part of the study.

Of these 102 responses, 45 (44%) indicated that the school had an emergency generic salbutamol inhaler available, while 57 (56%) indicated that the school did not. The proportion of schools in which emergency generic salbutamol inhalers were available varied by Local Authority from 0% to 71%.

## DISCUSSION

Within our study sample, 99% of mainstream secondary schools in North East England permitted pupils to carry their own inhaler medication: 77% without need for special permission or parental consent, and 22% with parental consent. Only 1 school (1%) stored inhalers brought in by pupils in a central location within the school, forbidding pupils from keeping them on their person.

Of the secondary schools in our study, 44% indicated that they held a generic salbutamol inhaler for use in an emergency.

### Strengths and weaknesses

To our knowledge this is the first study assessing the rules governing inhaler carriage in secondary schools within the UK and the first to assess secondary schools' uptake of an emergency salbutamol inhaler. This contributes data to a substantial gap in knowledge regarding the community-based clinical management of acute asthma exacerbations in children in the UK.

This data must, however, be carefully interpreted. The majority of our data collection was achieved via postal questionnaire. While this ensured a healthy response rate (69%), there is the possibility that the answers given do not reflect actual practice. Analysis of the medications policy for each school would have yielded more robust data; however, from our pilot study, few schools had a medications policy available on the school website making data collection via this method difficult. In addition, the school medication policy may be equally poorly reflective of actual practice. In addition, our questionnaire did not ask about policies in relation to any specific age group; we believe that this has limited impact on our results, as schools were asked to clarify in free text comments where policies varied from pre-described options, and several schools which had age-differentiated policies did so. An observational study would be ideal, but may be impractical.

All responses were independently scrutinised by both authors, and consensus reached in all cases. This limits the likelihood of observer bias in assessment of responses.

Data was collected over a seven month period from November 2014 to May 2015. Practice may have changed during the study period, especially in relation to whether the

schools held an emergency inhaler. Schools that responded to the postal questionnaire at the beginning of the research period may have subsequently changed their practice prior to the end of the study. Hence, our study is unlikely to be representative of practice at any single moment in time.

While the response rate was relatively high (69%), there is a substantial risk of response bias in this study. An unwillingness to respond to a postal questionnaire regarding asthma practice may plausibly be associated with poor practice in this area, such as not having a clear policy. At the individual level, it is well established that those who have not committed to a position on an issue are less likely to respond to questions about that issue (*Groves, Cialdini & Couper, 1992*); it is possible that the same is true of organisations such as schools.

The generalisability of the results is unclear. While coverage of secondary schools across a whole region of the UK is a strength of this study, there is no data to suggest whether similar practices are used in other parts of the UK. If, for example, Local Authorities took a collective approach to school medication policies in other parts of England, then practice may differ substantially. Further, the specific English legislative framework which sets the context for school policies in this area limits the international generalisability of the study.

## Comparison with other studies

The paucity of evidence in this area limits direct comparison with other studies.

The finding that 99% of included secondary schools in North East England allowed pupils to carry inhalers contrasts with a similar, but much older, study of primary schools in Birmingham in 1997 (*Evans & Kenkre, 1997*), where only 48% allowed pupils to carry inhalers. However, as discussed above, policies which may be rational regarding inhaler carriage in older children may be irresponsible when applied to younger children. Legislation surrounding carriage of inhalers in schools has also changed since this study was published.

The finding that 22% of included schools required parental permission for pupils to carry asthma inhalers bears further consideration against the literature on medical ethics. In England and Wales, the clear legal position is that children aged 16 or 17 years can assent to treatment, but that refusal of treatment in this age group can be overridden by a person with parental responsibility or a court of law. Yet, in certain schools, the described practice could possibly constrain pupils aged 16 years from independently assenting to prescribed, usually self-administered, medical treatment while on school grounds. This appears to represent something of an anomaly. Similar anomalies have been reported in the ethical literature: for example, children of this age being permitted to assent to treatment but not being permitted to sign the accompanying paperwork (*Terry, 2007*). Empirical ethical literature is less clear on the age at which children are capable of assenting to treatment (*Alderson, 1992*; *Ondrusek et al., 1998*), but clearly clinical practice is constrained by the contextual legal framework. It is notable that the legal framework in this area is inconsistent even within the constituent countries of the United Kingdom (*General Medical Council, 2007*), perhaps reflecting a high degree of ethical uncertainty.

In addition, it is notable that the legislation regarding use of emergency salbutamol inhalers in schools requires parental consent. This appears to place a legal limit on the

ability for schoolchildren aged 16 years or over to assent to this particular medical treatment. This curious legal anomaly has not yet been tested in court.

## Unanswered questions and future research

There is clear evidence that community-based treatment of acute asthma exacerbations in children is an important area for consideration in the wider picture of paediatric asthma cases (*NICE, 2013*). More research needs to be undertaken both regionally, nationally and internationally with the aim of highlighting areas in which asthma management can be improved for young people.

There are several unanswered questions raised by our study. Most importantly, our study used a process rather than outcome measure: it is unclear whether access to emergency asthma medication reduces the likelihood of adverse outcomes.

While our research gives an early indication of the proportion of schools who have purchased a generic emergency salbutamol inhaler, it is unclear whether the number of schools with a generic emergency salbutamol inhaler changed over the course of the study.

## Meaning of the study and implications

Our results provide the first research data on policies regarding carriage of inhalers in secondary schools in the UK, and an insight into schools' uptake of the emergency salbutamol inhaler. Further research is required to determine whether the results are replicated elsewhere. In addition, the level of variation in school policies demonstrated by this study may indicate a degree of variation in the quality of asthma care provided in schools. Standardisation to a policy based on robust evidence at a local, regional or national level may be desirable.

More consideration should be given to the ethical framework in which such studies exist. Current local policies and national legislation may unduly limit the right of 16-year-old pupils to assent to medical treatment in England and Wales.

### Funding
The authors received no funding for this work.

### Competing Interests
The authors declare there are no competing interests.

### Author Contributions
- Wendy Funston and Simon J. Howard conceived and designed the experiments, performed the experiments, analyzed the data, contributed reagents/materials/analysis tools, wrote the paper, prepared figures and/or tables, reviewed drafts of the paper.

### Data Availability
The raw data has been supplied as Data S1.

## Supplemental Information

Supplemental information for this article can be found online at http://dx.doi.org/10.7717/peerj.2006#supplemental-information.

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
