# Peer review of "A cross-sectional questionnaire study of the rules governing pupils' carriage of inhalers for asthma treatment in secondary schools in North East England"

_PeerJ, doi:10.7717/peerj.2006_

## Round 0.1 · original submission · Minor Revisions

Please carry out minor modifications as suggested by the reviewers.

Reviewer 1 ·

Basic reporting

Generally meets Peer J policies but there are some unreferenced statements, some clumsy grammar, some text in the wrong section, further clarification in parts, and more justification required for some of the conclusions

Experimental design

Acceptable

Validity of the findings

Mostly valid but some further clarification required on some aspects of interpretation.

Additional comments

Have tracked some comments and suggestions in attached copy of manuscript

Annotated reviews are not available for download in order to protect the identity of reviewers who chose to remain anonymous.

Reviewer 2 ·

Basic reporting

.

Experimental design

.

Validity of the findings

.

Additional comments

The interest of the article would seem to be school administrators, school nurses, head teachers and teachers; it is of limited interest outside the North East of England as an example of what other people might do to audit their current practice.

In this limited sense the novelty is fair – but maybe the significance of this is limited by the poor generalisability of the results.

The study is very simple and (mostly) clear. It would have been useful to have more information on which schools had and had not answered the questions and seen a table against the characteristics of the schools. Responses certainly seem to have been very variable by district. It would also be useful to know what the characteristics of the schools were according to whether they were compliant and noncompliant to current policies. It would be useful background information to know who in the school had completed the questionnaires (head teacher, administrator, school nurse). Finally (I couldn’t find the questionnaire) I was not clear whether the questionnaire specifically asked solely about policy in relation to students that were 16 years old and above. If not, and if the schools have a wider age range the answer to the question whether “students have to have parental permission to carry an inhaler” could be answered correctly by the same school either as “Yes (if they are under 16)” or “No, (if they are over sixteen they do not require parental approval).”

---

## Round 0.2 · accepted · Accept

Thank you for the comprehensive response to the reviewers. The manuscript is now accepted.